# Thirty Minutes Identified as the Threshold for Development of Pain in Low Back and Feet Regions, and Predictors of Intensity of Pain during 1-h Laboratory-Based Standing in Office Workers

**DOI:** 10.3390/ijerph19042221

**Published:** 2022-02-16

**Authors:** Beatriz Rodríguez-Romero, Michelle D. Smith, Sonia Pértega-Díaz, Alejandro Quintela-del-Rio, Venerina Johnston

**Affiliations:** 1Psychosocial Intervention and Functional Rehabilitation Research Group, Department of Physiotherapy, Medicine and Biomedical Sciences, Campus Oza, University of A Coruña, 15071 A Coruna, Spain; 2School of Health and Rehabilitation Sciences, The University of Queensland, Brisbane 4072, Australia; m.smith5@uq.edu.au (M.D.S.); v.johnston@uq.edu.au (V.J.); 3Rheumatology and Health Research Group, Department of Health Sciences, Campus Esteiro, University of A Coruña, 15471 Ferrol, Spain; s.pertega@udc.es; 4Department of Mathematics, University of A Coruña, 15071 A Coruna, Spain; alejandro.quintela@udc.es

**Keywords:** low-back pain, standing position, musculoskeletal pain, lower extremity

## Abstract

This study with 40 office workers investigated (a) the effect of time spent standing on low- back and lower limb pain during a 1-h laboratory-based task; (b) the standing time after which a significant increase in pain is likely; and (c) the individual, physical and psychosocial factors that predict pain. The primary outcome was bodily location of pain and pain intensity on a 100-mm Visual Analogue Scale recorded at baseline and every 15 min. Physical measures included trunk and hip motor control and endurance. Self-report history of pain, physical activity, psychosocial job characteristics, pain catastrophizing and general health status were collected. Univariate analysis and regression models were included. The prevalence of low-back pain increased from 15% to 40% after 30 min while feet pain increased to 25% from 0 at baseline. The intensity of low-back and lower limb pain also increased over time. A thirty-minute interval was identified as the threshold for the development and increase in low-back and feet pain. Modifiable factors were associated with low-back pain intensity—lower hip abductor muscle endurance and poorer physical health, and with feet symptoms—greater body mass index and less core stability.

## 1. Introduction

Office workers are known to adopt sedentary behaviors at work [1]. Although the available evidence has not confirmed a consistent causal relationship between occupational sitting and musculoskeletal pain [2,3,4], a growing body of evidence suggests that prolonged sitting is a major concern for the development of several chronic diseases, such as cardiovascular disease, diabetes type 2 and premature mortality [5]. Thus, it is no surprise that there is heightened interest in workplace initiatives to reduce the amount of sitting time for office workers [6,7] with alternatives such as sit-stand workstations or breaking up seated-work with standing-work. However, some concerns have been expressed that substituting sitting with standing may expose workers to new hazards and/or other health consequences. Two systematic reviews with meta-analysis in laboratory [8] and occupational [9] settings suggested that prolonged standing is associated with the occurrence of low-back and lower extremity symptoms, although the conclusions are inconclusive for the association with lower extremity symptoms. Similarly, causality between occupational standing and LBP has not been resolved, and not all people who are exposed to prolonged standing will develop LBP [2,10].

Experimental laboratory studies which simulate occupational standing have used an induced pain paradigm to identify factors which could be associated with developing low-back and/or lower extremity pain [8]. Some factors suggested to predispose a person to the development of LBP during prolonged standing are: (i) fatigue of the trunk and hip muscles [11,12,13]; (ii) movement control dysfunction [13,14,15,16]; and (iii) postural stiffness through increased levels of coactivation of hip and trunk muscles [14,17]. Specific factors associated with an increase in LBP during prolonged standing are unknown. Discomfort experienced in the feet and lower limbs during standing is often attributed to reductions in venous return and muscular fatigue [18]. The flowmetry, leg circumference, skin temperature, force through feet, and lower limb and trunk muscle activity have been reported as the main outcomes to study the possible mechanisms for lower limb symptoms; although the underlying mechanisms require further investigation [8,9,18].

Evidence for determining thresholds of excessive standing has not been elucidated. In a 2-year prospective study, Andersen et al. [19] demonstrated that standing at work for 30 min or more every hour elevated the odds for LBP by a factor of 2.1, and for pain in the hip, knee or foot by a factor of 1.7. A meta-analysis [9] suggested a statistically significant association between 2 or 4 h/workday of occupational standing and the occurrence of low back/lower extremity symptoms, although the authors highlighted that conclusions on the dose–response association cannot be drawn.

The aims of this laboratory study in office workers were to determine: (i) the effect of time spent standing on pain status during a 1-h laboratory-based standing task; (ii) the point after which significant increases in pain are likely; and (iii) the individual (e.g., age, sex, history of LBP, self-rated health), physical (e.g., deficits in motor control, muscle endurance) and psychosocial (e.g., job demands) factors that are associated with higher levels of low-back and lower limb pain after a 1-h standing task. Given the findings of previous studies, it was hypothesized that there would be a significant effect of time spent standing on the prevalence and intensity of low-back and lower limb pain, with a common threshold time when significant increases occur. We expect that specific individual (health status, BMI), physical (deficits in muscle endurance) and psychosocial (e.g., low job control) factors would be associated with higher scores of low-back and lower limb pain after a one-hour standing task.

## 2. Materials and Methods

### 2.1. Participants

A convenience sample of forty office workers, aged ≥ 18 years, who performed mostly sedentary work for ≥30 h per week were recruited. Participants were excluded if they: (i) were pregnant or less than six months postpartum, (ii) had any major trauma or surgery to the spine or lower limb in the last 12 months or (iii) had a diagnosis of neurological or systemic pathology. The recruitment process and sample size are described elsewhere [13].

The University of Queensland Human Research Ethics Committee B approved this study (Approval Number: #2017000666) and all participants provided informed consent prior to study participation. This study was registered in the Protocol Registration and Results System (PRS) (NCT03678623).

### 2.2. Study Procedure

Participants completed self-reported measures, undertook a physical examination conducted by a trained physiotherapist, and then participated in a 1-h standing task. These self-report measures were administered via an online survey completed the day prior or the same day as the laboratory testing session.

### 2.3. Measurements

Self-reported measures included (i) demographics; (ii) history of LBP (7-day prevalence); (iii) location of any bodily pain assessed with the Nordic Musculoskeletal Questionnaire [20] and pain assessed with a 100-mm Visual Analogue Scale (VAS) anchored with “no pain” at 0 and “worst pain imaginable” at 100 mm for each body location [21]; (iv) total and occupational physical activity assessed with the International Physical Activity Questionnaire (IPAQ) (MET-min/week) [22] and the Occupational Sitting and Physical Activity Questionnaire (OSPAQ) (minutes) [23], respectively; (v) psychosocial job characteristics evaluated through the Job Content Questionnaire (JCQ) which includes four domains (job control, psychological job demands, social support and physical demands) (4-point Likert) [24]; (vi) propensity for pain catastrophizing assessed with the Pain Catastrophizing Scale (PCS-total) (scores ranged from 0 to 52 with greater scores indicate a greater degree of catastrophizing) [25]; and (vii) general health status evaluated with the SF-12 through the Physical (PCS) and Mental (MCS) Component Summary scores [26].

At the start of the laboratory session, participants were given 15 min of seated rest while the testing protocol was explained. Physical testing was then undertaken in the following order: (i) height and weight; (ii) three motor control impairment tests: the active hip abduction test (AHAbd) (ranging from 0–5 as rated by participants and 0–3 by the examiner, with lower scores indicating better motor control for both ratings) [27], and the active straight leg raise test (ASLR) (ranging from 0–10 as the summed score of participant and examiner ratings, with lower scores indicating better motor control) [28]; (iii) endurance tests of the following trunk and hip muscles (measured as seconds able to holding a static position): abdominal endurance [29], supine bridge [30], isometric hip abduction [31] and Biering–Sorensen test [29]. The specific methodology on how each of these tests was applied is explained elsewhere [13].

The standing paradigm consisted of participants standing for an hour while performing their usual computer-based work. Participants stood within a rectangular floor space (122 × 61 cm) with their body fist-width away from the edge of a height-adjustable workstation. The workstation was standardized to each participant so that the desk height was 5–6 cm below the lateral epicondyle, the computer monitor was at arm’s length from the body, and the top of the computer monitor was at eye level. Participants were allowed to shift their weight as often as desired but were asked to keep both feet on the ground the majority of the time. The participant was not allowed to lean on the workstation with their arms, legs or trunk [13].

The primary outcome was pain status (yes/no) and severity of pain (VAS, 0–100 mm) in the low back and lower extremity (Figure 1). The workers self-reported the location of their pain on the body map and indicated pain intensity on the VAS at baseline, every 15 min during, and at the end of the standing test. The investigator verbally asked the participants to rate their pain. Participants were not given access to their previous scores.

### 2.4. Statistical Analysis

Descriptive statistics were computed for pain ratings over the 1-h task (0, 15, 30, 45 and 60 min). At each timepoint, prevalence of pain for each location was computed, and VAS scores were summarized as mean ± standard deviation and median.

For hip-thigh and knee-calf regions, the number of subjects who developed pain was small, and only descriptive analyses were performed. Therefore, only low-back and ankle-feet regions were analyzed in more detail. For those locations, an increment of ≥10 mm in VAS pain at any time between start and end of the test was considered to classify participants as Pain Developers (PD) or Non-Pain Developers (NPD) [9,32], and both groups were analyzed independently.

Since repeated pain evaluations were obtained for each subject at different timepoints, appropriate tests for paired data were employed in bivariate analysis. McNemar’s test for paired data was used to analyze the significance of changes in the prevalence of pain for each location during the 1-h task. In the same way, Wilcoxon’s signed-rank test was employed to compare pain scores between different timepoints.

Association of the different individual, physical and psychosocial factors recorded with pain ratings during the 1-h task was explored. First, the maximum change in VAS score from baseline and during the task was considered as the outcome using univariate and multivariate analyses. Spearman’s correlations were used to determine the strength of the relationship between each of the individual, physical and psychosocial factors included and the increase in low-back and ankle-feet pain during the task. Then, a stepwise multivariate linear regression model was adjusted, including as covariates those with the highest associated correlations in the univariate analysis, both for the whole group of workers and for the PD group. Finally, in order to compare the consistency of the results, univariate and multivariate linear mixed-effects random-slope repeated measures models were also adjusted [33]. This type of model assumes that time effects (changes in pain rating over time) are random among individuals, considering the correlation among repeated measures in the same subject. Regression coefficients were estimated for the interaction between each of the covariates and time, allowing the rate of change to vary for different baseline characteristics.

Statistical analyses were performed using software SPSS version 25.0 (SPSS Inc., Chicago, IL, USA) and R version 4.0.5 (R Foundation for Statistical Computing, Vienna, Austria), with a bilateral significance level set at *p* < 0.05.

## 3. Results

Forty office workers (22 females; mean age: 37.4 ± 6.6 years; BMI: 26.3 ± 5.7 and 58% considered within healthy weight range) were included in the study. All participants completed the laboratory testing (physical testing and 1-h standing task) with no adverse events reported.

### 3.1. Standing-Time Effect on Pain Status: Any Reported Pain

The number of workers who reported pain in the low back and lower limb, throughout the task, increased over time (Table 1). At the beginning of the standing task (0 min), 15% of the participants reported some degree of LBP, increasing up to 30% at 15 min (*p* = 0.070) and reaching 40% (*p* = 0.006) and 42.5% (*p* = 0.003) at 30 and 45 min, respectively (Figure 2a). None of the participants had ankle-feet pain at the beginning of the task. Prevalence of ankle-feet pain was 10% after 15 min, increasing to 25% at 30 min (*p* = 0.031) and reaching 35.0% both at 45 min (*p* = 0.006) and 60 min (*p* = 0.002) (Figure 2b). Low-back and ankle-feet pain prevalence did not significantly increase after 30 min. While there was an increase in the number of participants who reported lower limb pain between baseline and 60 min (2.5% to 15% for the hip-thigh region and 5% to 27.5% for the knee-calf), this did not reach statistical significance.

### 3.2. Standing-Time Effect on Pain Status: Intensity of Pain

Of the 40 participants, 14 office workers were considered low-back pain developers and 9 were ankle-feet-pain developers (abbreviated as ankle-feet-PD onwards). For the hip-thigh and knee-calf regions, the number of participants who reported having a change of ≥10 on the VAS was small (3 and 6, respectively) preventing analysis.

The raw VAS score and VAS score increased over time for the total sample, PD and NPD groups, both for low-back and ankle-feet regions, are shown in Table 2 and Appendix A There was a significant standing-time effect, with individuals identified as PD showing increased levels of pain over time and the NPD group remaining at a very low level. The low-back-PD group averaged a mean VAS score of 30.8 ± 20.5 mm and the ankle-feet-PD group averaged a mean VAS score of 22.6 ± 9.7 mm at the end of standing. In addition, these results show that after 30 min of standing, significant differences in pain scores appear from the baseline.

### 3.3. Association of Individual, Physical and Psychosocial Factors with Pain Ratings over Time

Table 3 shows the Spearman’s rho correlation coefficients between the maximum increment of VAS scores for the low back and ankle-feet during the task and different individual, physical and psychosocial factors in the whole sample. Correlations for PD and NPD groups are shown in Appendix A.

Considering the whole sample, the maximum increase in VAS scores at the low back throughout of the 1-h standing task was significantly correlated with (i) history of LBP measured as the severity of LBP in the last 7 days (Rho = 0.54, *p* = 0.000); (ii) SF-12 (physical component summary) (Rho = −0.35, *p* = 0.029); (iii) ASLR, total participant-score (Rho =0.35, *p* = 0.029); (iv) AHAbd left side, participant-score (Rho = 0.33, *p* = 0.04); (v) isometric hip abduction (right leg) (Rho = −0.47, *p* = 0.002); (vi) isometric hip abduction (left leg) (Rho = −0.48, *p* = 0.002). In the feet area, the maximum increase in VAS scores was significantly correlated with (i) BMI (Rho = 0.38, *p* = 0.016); (ii) SF-12 (mental component summary) (Rho = 0.39, *p* = 0.014); (iii) Supine Bridge (Rho = −0.33, *p* = 0.04); and (iv) Biering–Sorensen test (Rho = −0.32, *p* = 0.048).

### 3.4. Regression Analysis-Predicting the Magnitude of Low-Back Pain throughout of 1-h Standing Task

A multivariate linear regression model was adjusted with the maximum increment in VAS scores throughout of standing as the dependent variable and the magnitude of the highest and significant correlated variables (determined using Spearman’s correlation) as covariates (Table 4). Lower hip abductor muscle endurance (B = −0.23, *p* = 0.007) and lower physical health (PCS-SF-12) (B = −0.86, *p* = 0.008) predicted higher level of LBP throughout of the 1-h standing task for the entire sample. These two variables explained 41% of the variability in the increment of the pain (R^2^ = 0.41).

The mixed regression model reinforced these results, with the interaction time × minimum isometric hip abduction and time×PCS-SF-12 being significant predictors of the highest level of LBP during the task. These results indicate that workers with lower hip abductor muscle endurance and lower physical health experience a higher rate of increase of LBP during the task.

## 4. Discussion

This study showed significant time-based changes for the reported prevalence and severity of pain in the low back and ankle-feet regions of the body, with 30 min identified as the threshold for observing these differences. The regression models demonstrated that less hip abductor muscle endurance and less physical health (as measured with the SF-12) predicted a greater increase of LBP at the end of a 1-h standing task. The correlation analysis suggested that the maximum increase in VAS score in the ankle-feet area was associated with higher BMI, less back and hip muscle endurance (Supine Bridge and Biering–Sorensen test) and mental health (SF-12).

There was a significant effect of standing at a workstation for 1-h on the presence and intensity of pain in the low-back and ankle-feet regions. This has consistently been reported for the low back [9,11,14,17,34], but has been less commonly investigated in the ankle-feet region [9,18,35]. The percentage of office workers who developed LBP at the end of the 1-h standing task (42.5%) was similar to the average of 44% reported in the systematic review of Coenen et al. [8]; but lower than other studies which reported rates of 81% [34], 71% [11] and 65% [17]. The VAS level of pain reported by low-back PD in this study was within the range of what has been reported previously (19 mm [34] to 32 mm [11]), despite most previous work using populations of students without a history of LBP and using a longer standing task. The average increase in pain scores from baseline in our study was 27.9 ± 18.1 mm which is lower than clinical low back populations but higher than the absolute cut-off value of 15 points for the minimal important change (MIC) on the VAS for LBP patients [36].

Lower limb pain was commonly reported during the standing task by our participants, with 15% reporting hip-thigh pain, 27.5% reporting pain in the knee-calf region and 35% developing ankle-feet pain at the end of standing. It should be noted that no participants had ankle-feet pain at the beginning of the standing task. The increase in ankle-feet-PD pain was 22.6 ± 21 mm. Although this level of pain could be interpreted as low, it is higher than the MIC (9.3 mm) reported for the clinical interpretation of results in patients with foot or ankle pathologies [37]. This pain intensity for the ankle-feet region is similar to that reported by Antle and Cote [18] after 34 min of standing (3.5 out of 10) and Smith et al. [35] after 2 h of standing (1.8 out of 10).

Our results demonstrate that 30 min of standing affects the prevalence and severity of pain at the low-back and ankle-feet regions in office workers. After 30 min, the PD group reported pain scores which exceeded the MIC [36,37]. It has been demonstrated that in PD low back usually begins to develop within 30–60 min of standing [11,14,17,35]. Prolonged periods of occupational standing (greater than 30 min each hour), was one of the strongest predictors of LPB in employees [19] and low-back symptoms were reached after 71 min with this reduced to 42 min in those considered PD, after pooled data from laboratory studies [8]. Dose–response associations for lower extremity symptoms are more heterogeneous in the literature [8], with one study showing lower limb discomfort after 34 min of standing [18]. While a significant interaction of time on pain development during standing has been illustrated [11,14,17], most studies did not determine the point of time when the significant increase in pain occurred. Consequently, based on our results, we recommend that office workers restrict their maximum standing time to no more than 30 min which is less than the suggested time of 40 min [8].

Few studies have considered the predictors of pain intensity during standing. Multivariate regression modelling demonstrated that lower hip abductor muscle endurance and lower physical health were independently associated with a higher level of LBP, explaining 41% of the variability in pain in our study. In turn, mixed regression modelling demonstrated a significant interaction for time×isometric hip abduction and time×SF-12 (physical component summary), reinforcing the importance of these two factors as predictors of the highest-level LBP during the 1-h standing task. Hip abductor muscle endurance has previously suggested to be associated with LBP development during standing. Viggiani and Callaghan [12] determined that hip abductor fatigability (measured using isometric hip abduction) differentiated those PD from NPD, with PD having lower hip abductor endurance. Marshall et al. [11] showed that low-back PD had lower gluteus medius endurance (measured with a side-bridge test), there was no association between the side bridge test and pain levels in linear regression analysis. Finally, Hwang et al. [38] identified that hip abductor muscle strength (measured in the similar way as this study) was the variable that most contributed to VAS scores in workers with LBP who performed occupational standing. While causality between LBP and hip abductor muscle function cannot be confirmed from existing research, the negative slope in our analysis indicates that low-back VAS scores decrease as endurance of the hip abductors muscles increases. Together, data suggest that hip abduction muscle weakness and fatiguability may be important to consider in preventing or managing standing-induced LBP.

To our knowledge, this is the first study to investigate self-perceived physical health, measured through Physical Component Summary (SF-12), as predictor of higher pain intensity during a prolonged standing task. In the calculation of the PCS summary score, the highest weights were given to four domains: physical functioning, role physical, bodily pain, and general health. Bodily Pain was the domain with the lowest scores for our sample (mean: 49.6, IQR: 46.9–57.5, data not shown previously) and for the LBP-PD group (47.4 ± 10.5). This is consistent with previous reports stating that the presence of pain (in the low back and/or other sites) may be associated with the incidence and prevalence of LBP [39,40].

Factors associated with ankle-feet pain have not previously been investigated in relation to occupational standing in office workers. While regression analysis was not undertaken for the maximum VAS for ankle-feet pain due to small proportion of participants with this pain, ankle-feet region VAS was moderately correlated with greater BMI, lower trunk/hip muscle function (lower supine bridge and Biering–Sorensen test ability) and mental health. There is a strong association between high BMI and non-specific foot pain in the general population [41] and in people with plantar heel pain [42]. Mechanical loading with increased body mass is a possible mechanism for this relationship, as well as metabolic and psychological factors [41]. Trunk/hip stability is thought to be important for the production, transfer and control of force throughout the entire kinetic chain [43]. Gluteus maximus and medius muscles weakness has been identified in people with ankle injuries [44,45], and it has been hypothesized that deficits in core stability may increase risk of lower extremity injury [46,47]. Altered loading throughout the lower kinetic due to hip/trunk muscle deficits may contribute to ankle-feet pain during prolonged standing at a standing workstation. The association between better mental health and ankle-feet pain could be due to a spurious finding and a consequence of the small sample size.

There are limitations to this study that must be considered. First, the study has a relatively small sample size (*n* = 40), with only a small proportion of subjects classified as low-back or ankle-feet PD. Although previous studies have had similar sample sizes [12,14], this sample size may have led to two important issues: (i) failing to detect a real effect on pain development for any of the variables studied, and (ii) finding effects that seem supported by the data but are spurious. The use of alternative statistical methods would have been preferred [48] but were difficult due to the small sample. The results of this study should be confirmed in future studies involving more individuals and more analysis techniques. Second, this study did not include outcomes to identify potential vascular mechanisms that may be associated with the development of musculoskeletal ankle-feet symptoms [8,18]. We recommend that future research include such measures.

This study has shown that there is an impact of 1 h-standing exposure on several aspects of pain status in office workers, determined the significant dose–response relationship for standing, and clarified the factors associated with the intensity of low-back and ankle-feet pain. Based on the findings of our study, practitioners and clinicians should advise office workers to avoid standing for more than 30 min in light of the dose–response relationship for standing and pain. Due to the relationship between hip abductor muscle endurance, physical health status and intensity of LBP, future research is needed to determine if improving these factors decreases LBP intensity during standing. Similarly, further investigation is needed to understand the relationship between BMI, trunk/hip muscle function and mental health on ankle-feet pain during standing.

## 5. Conclusions

This study in office workers demonstrated that the prevalence and intensity of low- back and ankle-feet pain increased during a 1-h laboratory-based standing task, with 30 min identified as the threshold for the development/provocation of pain. Lower hip abductor muscle endurance and physical health predicted the low-back pain intensity. In the ankle-feet area, results suggest that the increase in pain scores was correlated with greater BMI and lower trunk/hip muscle function.

## Figures and Tables

**Figure 1 ijerph-19-02221-f001:**
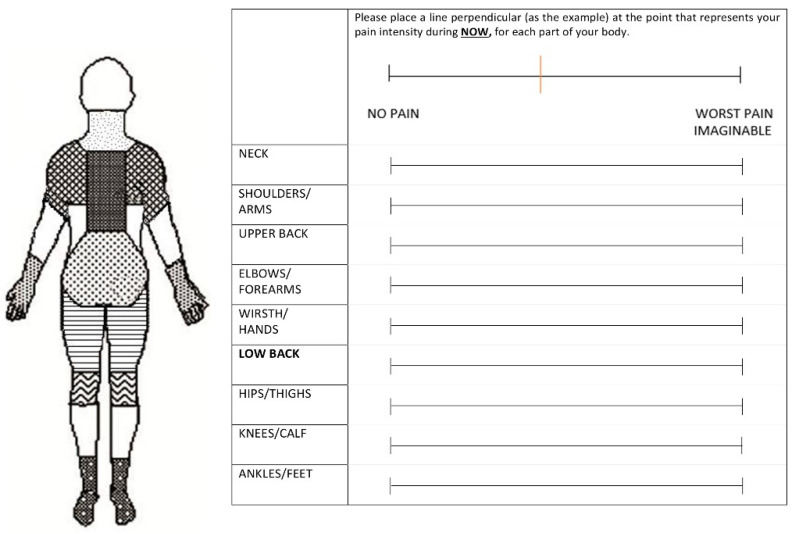
Body Pain Diagram and Visual Analogue Scale used every 15 min for reporting location and intensity of pain during the standing task.

**Figure 2 ijerph-19-02221-f002:**
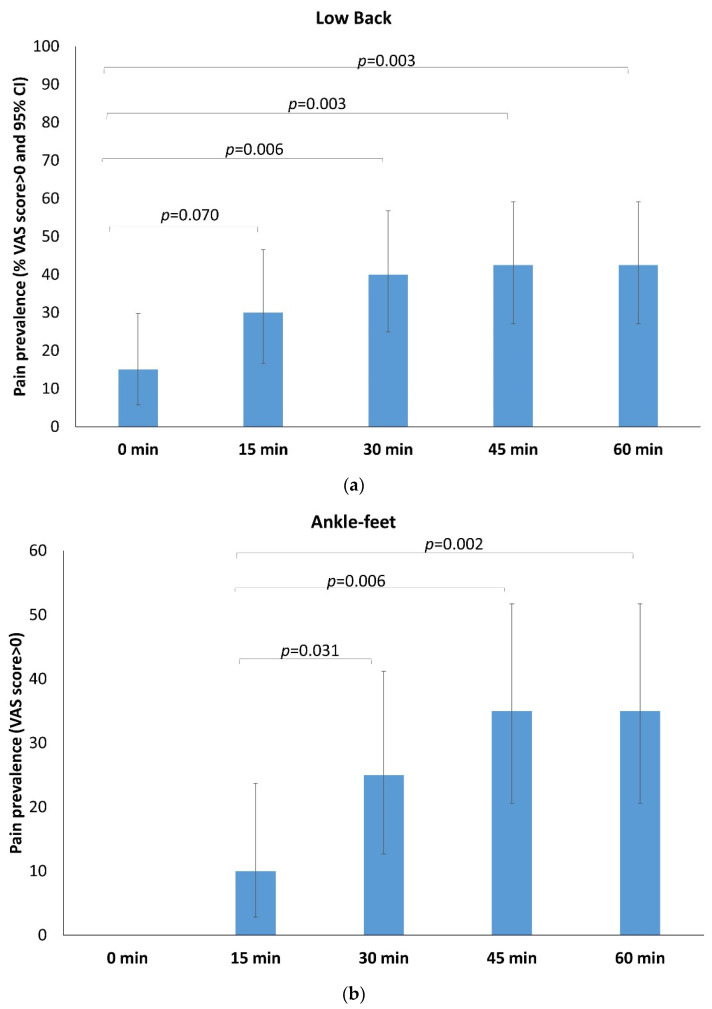
Prevalence and time-based changes of any reported pain (VAS scores > 0) for each area: (**a**) for low back; (**b**) for ankle-feet. *p*: McNemar test.

**Table 1 ijerph-19-02221-t001:** Prevalence of any low-back and lower extremity pain at 0, 15, 30, 45 and 60 min throughout the 1-h standing task.

**Low Back**										
Pain	0 min		15 min		30 min		45 min		60 min	
	*n*	%	*n*	%	*n*	%	*n*	%	*n*	%
No	34	85.0	28	70.0	24	60.0	23	57.5	23	57.5
Yes	6	15.0	12	30.0	16	40.0	17	42.5	17	42.5
Total	40	100.0	40	100.0	40	100.0	40	100.0	40	100.0
**Hip-Thigh**										
Pain	0 min		15 min		30 min		45 min		60 min	
	*n*	%	*n*	%	*n*		*n*	%	*n*	%
No	39	97.5	39	97.5	35	87.5	34	85.0	34	85.0
Yes	1	2.5	1	2.5	5	12.5	6	15.0	6	15.0
Total	40	100.0	40	100.0	40	100.0	40	100.0	40	100.0
**Knee-Calf**										
Pain	0 min		15 min		30 min		45 min		60 min	
	*n*	%	*n*	%	*n*		*n*	%	*n*	%
No	38	95.0	36	90.0	33	82.5	30	75.0	29	72.5
Yes	2	5.0	4	10.0	7	17.5	10	25.0	11	27.5
Total	40	100.0	40	100.0	40	100.0	40	100.0	40	100.0
**Ankle-feet**										
Pain	0 min		15 min		30 min		45 min		60 min	
	*n*	%	*n*	%	*n*	%	*n*	%	*n*	%
No	40	100	36	90.0	30	75.0	26	65.0	26	65.0
Yes	-	-	4	10.0	10	25.0	14	35.0	14	35.0
Total	40	100.0	40	100.0	40	100.0	40	100.0	40	100.0

**Table 2 ijerph-19-02221-t002:** VAS scores (0–100) and change in VAS scores from baseline at the low-back and ankle-feet regions at 0, 15, 30, 45 and 60 min of the 1 h standing task for all study participants, PD and NPD groups.

	Low Back				
	VAS Scores (0–100 mm)		VAS Scores Increase from Baseline (0–100 mm) ^#^		
	Mean ± SD	Median	Mean ± SD	Median	*p* *
Total (*n* = 40)					
0 min	1.63 ± 4.5	0	-	-	
15 min	3.57 ± 8.7	0	1.94 ± 6.4	0	0.050
30 min	7.54 ± 13.2	0	5.90 ± 11.8	0	0.004
45 min	10.01 ± 17.6	0	8.38 ± 16.4	0	0.002
60 min	11.64 ± 19.0	0	9.97 ± 17.5	0	0.001
PD (*n* = 14)					
0 min	2.9 ± 5.9	0.0	-	-	-
15 min	8.7 ± 13.3	4.0	5.9 ± 8.1	3.0	0.050
30 min	20.5 ± 15.5	19.0	17.6 ± 12.7	19.0	0.004
45 min	27.0 ± 21.1	18.0	24.1 ± 19.0	18.0	0.002
60 min	30.8 ± 20.5	23.5	27.9 ± 18.1	23.5	0.001
NPD (*n* = 26)					
0 min	1.0 ± 3.5	0.0	-	-	-
15 min	0.8 ± 2.1	0.0	−0.2 ± 4.0	0.0	0.999
30 min	0.6 ± 1.6	0.0	−0.4 ± 3.9	0.0	0.684
45 min	0.9 ± 2.1	0.0	−0.1 ± 4.1	0.0	0.916
60 min	0.9 ± 3.1	0.0	−0.1 ± 3.6	0.0	0.715
	**Ankle-feet**				
Total (*n* = 40)					
0 min	0.0 ± 0.0	0	-	-	
15 min	1.5 ± 5.4	0	1.5 ± 5.4	0.0	0.068
30 min	3.2 ± 6.3	0	3.2 ± 6.3	0.0	0.005
45 min	5.0 ± 7.9	0	5.0 ± 7.9	0.0	0.001
60 min	5.8 ± 70.4	0	5.8 ± 10.4	0.0	0.001
PD (*n* = 9)					
0 min	0.0 ± 0.0	0.0	-	-	-
15 min	6.8 ± 10.1	0.0	6.8 ± 10.1	0.0	0.068
30 min	11.6 ± 7.2	12.0	11.6 ± 7.2	12.0	0.012
45 min	15.7 ± 4.6	17.0	15.7 ± 4.6	17.0	0.008
60 min	22.6 ± 9.7	21.0	22.6 ± 9.7	21.0	0.008
NPD (*n* = 31)					
0 min	0.0 ± 0.0	0.0	-	-	-
15 min	0.0 ± 0.0	0.0	0.0 ± 0.0	0.0	0.999
30 min	0.8 ± 3.3	0.0	0.8 ± 3.3	0.0	0.180
45 min	1.8 ± 5.5	0.0	1.8 ± 5.5	0.0	0.027
60 min	0.9 ± 2.3	0.0	0.9 ± 2.3	0.0	0.026

* *p*-values from two-sided Wilcoxon’s signed-rank test comparing VAS score at each moment relative to baseline VAS pain. **^#^** Positive scores indicate an increase in VAS; PD: pain developers; NPD: non-pain developers.

**Table 3 ijerph-19-02221-t003:** Spearman correlation coefficients between the maximum increment at VAS scores at low- back and ankle-feet regions throughout of the standing task and variables for the entire sample (*n* = 40).

	Low Back		Ankle-Feet	
	Total *n* = 40		Total *n* = 40	
	Rho	*p*	Rho	*p*
Age (years)	0.162	0.318	−0.008	0.961
BMI (kg/m^2^)	−0.098	0.546	0.379 *	0.016
IPAQ, during de last 7 days, how much time did you usually spend sitting on a weekend day (minutes)	−0.002	0.988	0.284	0.076
IPAQ, during de last 7 days, how much time did you usually spend sitting on a weekday (minutes)	0.142	0.381	0.025	0.878
IPAQ, MET min/week	0.005	0.975	0.067	0.690
LBP severity, last 7 days (0–100)	0.538 *	0.000	−0.046	0.777
OSPAQ, minutes sitting at work per week	−0.268	0.094	0.283	0.077
OSPAQ, minutes standing at work per week	0.028	0.862	−0.009	0.956
OSPAQ, minutes walking at work per week	−0.083	0.612	−0.208	0.198
JCQ, Job Control	−0.085	0.601	−0.068	0.679
JCQ, Psychological Job Demands	−0.140	0.390	−0.220	0.172
JCQ, Social Support	0.009	0.957	0.104	0.523
JCQ, Physical Demands	0.158	0.329	0.010	0.953
PCS, Rumination	0.222	0.168	−0.121	0.455
PCS, Magnification	0.141	0.386	−0.224	0.164
PCS, Helplessness	0.263	0.101	0.041	0.801
PCS-total	0.249	0.122	−0.121	0.458
SF-12, Mental Component Summary	0.278	0.083	−0.387 *	0.014
SF-12, Physical Component Summary	−0.345 *	0.029	0.168	0.299
ASLR, total examiner-score (0–10)	0.210	0.193	0.051	0.755
ASLR, total participant-score (0–10)	0.346 *	0.029	−0.058	0.724
AHAbd, right side, examiner-score (0–3)	0.038	0.816	0.025	0.881
AHAbd, left side, examiner-score (0–3)	0.129	0.426	0.033	0.839
AHAbd, right side, participant-score (0–5)	0.226	0.162	0.115	0.479
AHAbd, left side, participant-score (0–5)	0.325 *	0.041	0.008	0.960
Abdominal (s)	−0.269	0.094	−0.156	0.337
Side Bridge right side (s)	−0.254	0.114	−0.097	0.550
Side Bridge left side (s)	−0.246	0.126	−0.069	0.671
Supine Bridge (s)	−0.298	0.062	−0.327 *	0.040
Isometric hip abduction (right leg) (s)	−0.472 *	0.002	−0.003	0.985
Isometric hip abduction (left leg) (s)	−0.484 *	0.002	0.045	0.782
Sorensen (s)	−0.290	0.070	−0.315 *	0.048

* *p* ≤ 0.05; BMI, Body Mass Index; MET, Metabolic Equivalent of Task (computed as the sum of walking, moderate-intensity, and vigorous-intensity physical activity); OSPAQ, Occupational Sitting and Physical Activity Questionnaire; JCQ, Job Content Questionnaire; ASLR, Active Straight Leg Raise; AHAbd, Active Hip Abduction; s, seconds.

**Table 4 ijerph-19-02221-t004:** Multivariate lineal and mixed regression models for the prediction of low-back pain throughout of the 1-h standing task.

Lineal Regression Analysis					
Model	B	SE	*p*	95% CI	
				Lower	Upper
Intercept	74.661	15.375	0.000	43.507	105.814
Isometric Hip Abduction endurance test	−0.233	0.081	0.007	−0.397	−0.069
Physical Component Summary (SF-12)	−0.864	0.309	0.008	−1.489	−0.239
**Mixed Regression Analysis**					
	B	SE	*p*		
Fixed effects					
Intercept	−8.07	6.31	0.203		
Time	0.79	0.42	0.068		
Age	0.076	0.076	0.322		
Time×Age	0.009	0.005	0.078		
Isometric Hip Abduction endurance test	−0.002	0.025	0.939		
Time*Isometric Hip Abduction endurance test	−0.004	0.002	0.022		
Physical Component Summary (SF-12)	0.132	0.098	0.177		
Time*Physical Component Summary (SF-12)	−0.012	0.006	0.062		
	Estimate	SE			
Random effects					
Linear slope (time)	0.073	0.270			
Residual	32.30	5.68			

SE: Standard error; CI: Confidence Interval.

## Data Availability

https://zenodo.org/record/5947650#.YfwuJviCFhF.

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
