# Peer review of "Thirty Minutes Identified as the Threshold for Development of Pain in Low Back and Feet Regions, and Predictors of Intensity of Pain during 1-h Laboratory-Based Standing in Office Workers"

_ijerph, 2022, doi:10.3390/ijerph19042221_

Round 1

Reviewer 1 Report

The number of subjects in this study is relatively small (40), but as the authors state, the results seem to be reasonably consistent with previous data.

There are, however, a few points of concern.

With regard to the experimental method, there is no clear description of the condition of the test subjects before the start of standing experiment. It would seem that a preparation period of 30 minutes of seated rest would be required.

The highlight of the study is probably the increase in the prevalence of low back pain and ankle-foot pain in Figure 2, but McNemar's test does not seem to be the best test in this case.
In the first place, the reason why the threshold for low back pain is set at 30 minutes is, I suppose,  because this is the time at which a comparison test with 0 minutes shows a 5% significance, but this is due to the fact that the experiment was conducted with a small number of people (40), and if the experiment were conducted with 1000 people, the difference would be clearly significant at 15 minutes. Therefore, it is inappropriate to use it as a threshold definition. (Rather, it seems that the choice of McNemar's test method is erroneous, as the figure shows a significant increase in low-back pain at 15 minutes).
The reason why controls in Figure 2a and 2b are different (at 0 and 15 minutes, respectively) is probably due to choose the use of McNemar's test erroneously.

One point that may be of interest to the reader is that, as can be seen in Figure 2, both low-back pain and ankle-foot pain seem to be almost saturated at 1 hour and do not seem to increase any further, which could be used as a means of screening test for PD if standardised, and that even at 30 minutes it have shown that it is possible to find about 90% of PD at 30 minutes. (In this sense, the test of the significant difference between 1 hour and 30 minutes may be more meaningful?)

As PD (pain developer), authors included those who had low back pain before standing work, but it might have been clearer to exclude those who originally had non-zero scores for back pain etc. in the experimental design.

There are other minor points of concern, but I hope that authors will reconsider some of the important ones that relate to the main conclusion (definition of the threshold).

Author Response

Thank you for the opportunity to resubmit this paper. 

We provide a point-by-point response to the reviewer´s comments in the attach.

Reviewer 2 Report

Many thanks for asking me to review this article. I really appreciate the concept of this article. The authors have undertaken an huge effort to study the correlation between time threshold and development of pain in low back and feet regions. However, before the paper could be published, several revisions/edits would have to be made. It is especially important to correct citations and simplify tables, making them more readable. The manuscript should be more carefully formatted and language should be enhanced.

Please consider including more details about statistical analysis. Why authors use specific statistical tests – I didn’t see an explanation.

Please kindly consider increasing sample size in the study - do authors have any possibility to extend the group or compare to another? 

Could authors improve citations in the manuscript as follows?

Section 2.3.

Line 136. Please correct reference.

Figure 1 should be placed in better resolution.

Section 3.1.

Line 187, 190, 193. Please correct references.

Section 3.2.

Line 187, 190, 193. Please correct references.

Table 2 – authors should consider simplifying the table.

Section 3.3.

Line 228, 231. Please correct references. 

Section 3.4.

Line 257. Please correct reference.

Multiple references should be formatted such as [35-38] or this [34,35,38].

Section 5. Line 470. Please complete the Data Availability Statement.

I hope you find my comments helpful.

Kind regards

Author Response

(The authors gave the same response as above.)

Reviewer 3 Report

Dear Authors

I reviewed the manuscript # ijerph-1555146 entitled "Thirty minutes identified as the threshold for development of pain in low back and feet regions, and predictors of intensity of pain during 1-hour laboratory-based standing in office workers." which you submitted to the International Journal of Environmental Research and Public Health (IJERPH).

The title and whole text of this study seem to be consistent with the International Journal of Environmental Research and Public Health (IJERPH). I think this paper will be better if some minor and major points are corrected.

Minor points

Do not put '.' in the title.

Whole Text 1: All 'p' used as statistical symbols should be changed to italics.

Whole Text 2: When using multiple references, reference numbers should not be used separately.

Line 34: [2][3][4] → [2-4]

Line 38: [6][7] → [6,7]

Line 47: [2][10] → [2,10]

Line 52: [11][12][13] → [11-13]

Line 52: [14][15][16][13] → [13-16]

Line 53: [17][14] → [14,17]

Line 59: [8][9][18] → [8,9,18]

Line 154: [9][33] → [9,33]

Line 288: [35][14][17][11][9] → [9,11,14,17,35]

Line 292: [14][8] → [8,14]

Line 298: [18][8] → [8,18]

Line 323: [36][37] → [36,37]

Line 325: [14][17][11] → [11,14,17]

Line 331: [14][17][11] → [11,14,17]

Line 385: [39][40] → [39,40]

Line 401: [8][18] → [8,18]

Line 408: [44][45] → [44,45]

Line 409: [46][47] → [46,47]

Line 420: [12][14] → [12,14]

Line 428: [8][18] → [8,18]

Whole Text 3: All 'n' indicating the number of subjects should be changed to italics.

Whole Text 4: When using all abbreviations for the first time, such as LBP on Line 45, use the full name (abbreviation), and the abbreviation must be presented thereafter.

Line 136: .... (Error! Reference source not found.) appears frequently in the text. It must be corrected.

Line 181: There must be a space between all numbers and symbols (±).

Line 189: There must be a space between all the statistical symbols 'p' and the equal sign (=).

In Table 2, '.' is marked with ',' in many cases. Correction is required.

Major points

Line 87: The average age, average height, average weight, and the number of months worked of the study subjects should be presented.

The authors described in Line 83 as "Participants were excluded ..... last 12 months or had a diagnosis of neurological or systemic pathology." However, looking at Table 1, it is indicated that 6 people have pain at 0 min of 'Low Back' in the 1h-standing task. And 1 person at 0 min of 'Hip-Thigh' and 2 people at 0 min of 'Knee-Calf' is indicated as having pain. Shouldn't these subjects be excluded from the study?

Figure 2 is too crude. The numbers in it are hard to see, and the 'line' to show the 'difference' must also rise to the top of the standard deviation. Since the standard deviation values (±) are the same, it is okay to show only the '+' value.

The 'discussion' of this paper is very awkward. Do you need to divide the discussion into several sections? If you have to divide it when you look at the title, there are two large sections, so dividing it into two sections can enhance the reader's understanding.

Sincerely,

Author Response

(The authors gave the same response as above.)

Round 2

Reviewer 2 Report

I'm satisfied with the introduced corrections. As far as I am concerned I have only little comment mentioned below. Good work!

Please fix the issue "Error! Reference source not found." which I have found in line: 132, 190, 193, 196, 219, 234, 237, 263, before publication of the manuscript.

Reviewer 3 Report

Review Letters

Dear Authors

I re-reviewed the manuscript # ijerph-1555146 entitled "Thirty minutes identified as the threshold for development of pain in low back and feet regions, and predictors of intensity of pain during 1-hour laboratory-based standing in office workers." which you submitted to the International Journal of Environmental Research and Public Health (IJERPH). I think this paper was better than old version. However, there are some minor points in the Figure 2.

Minor points

The size of the text shown in the Figure 2 is too small. I would like to increase the font size. Also, it would be good to change the 'p' to italics. Finally, if the shape and color of the graph of the Figure 2 are also refined, I think it will increase the understanding of the readers.

Sincerely,
